# Possible Transmission Flow of SARS-CoV-2 Based on ACE2 Features

**DOI:** 10.3390/molecules25245906

**Published:** 2020-12-13

**Authors:** Sk. Sarif Hassan, Shinjini Ghosh, Diksha Attrish, Pabitra Pal Choudhury, Alaa A. A. Aljabali, Bruce D. Uhal, Kenneth Lundstrom, Nima Rezaei, Vladimir N. Uversky, Murat Seyran, Damiano Pizzol, Parise Adadi, Antonio Soares, Tarek Mohamed Abd El-Aziz, Ramesh Kandimalla, Murtaza M. Tambuwala, Gajendra Kumar Azad, Samendra P. Sherchan, Wagner Baetas-da-Cruz, Kazuo Takayama, Ángel Serrano-Aroca, Gaurav Chauhan, Giorgio Palu, Adam M. Brufsky

**Affiliations:** 1Department of Mathematics, Pingla Thana Mahavidyalaya, Maligram 721140, India; sarimif@gmail.com; 2Department of Biophysics, Molecular Biology and Bioinformatics, University of Calcutta, Kolkata 700009, India; shinjinighosh2014@gmail.com; 3Dr. B. R. Ambedkar Centre for Biomedical Research (ACBR), University of Delhi (North Campus), Delhi 110007, India; dikshaattrish@gmail.com; 4Applied Statistics Unit, Indian Statistical Institute, Kolkata 700108, West Bengal, India; pabitra@isical.ac.in; 5Department of Pharmaceutics and Pharmaceutical Technology, Yarmouk University-Faculty of Pharmacy, Irbid 566, Jordan; alaaj@yu.edu.jo; 6Department of Physiology, Michigan State University, East Lansing, MI 48824, USA; bduhal@gmail.com; 7PanTherapeutics, Rte de Lavaux 49, CH1095 Lutry, Switzerland; lundstromkenneth@gmail.com; 8Research Center for Immunodeficiencies, Pediatrics Center of Excellence, Children’s Medical Center, Tehran University of Medical Sciences, Tehran 1416753955, Iran; rezaei_nima@tums.ac.ir; 9Network of Immunity in Infection, Malignancy and Autoimmunity (NIIMA), Universal Scientific Education and Research Network (USERN), SE-123 Stockholm, Sweden; 10Department of Molecular Medicine, Morsani College of Medicine, University of South Florida, Tampa, FL 33612, USA; 11Doctoral studies in natural and technical sciences (SPL 44), University of Vienna, 1010 Wien, Austria; muratseyran@gmail.com; 12Italian Agency for Development Cooperation—Khartoum, Sudan Street 33, Al Amarat, Khartoum 825109, Sudan; damianopizzol8@gmail.com; 13Department of Food Science, University of Otago, Dunedin 9054, New Zealand; parise.adadi@postgrad.otago.ac.nz; 14Department of Cellular and Integrative Physiology, University of Texas Health Science Center at San Antonio, 7703 Floyd Curl Dr, San Antonio, TX 77030, USA; soaresa@uthscsa.edu (A.S.); mohamedt1@uthscsa.edu (T.M.A.E.-A.); 15Zoology Department, Faculty of Science, Minia University, El-Minia 61519, Egypt; 16Applied Biology, CSIR-Indian Institute of Chemical Technology Uppal Road, Tarnaka, Hyderabad 500007, India; ramesh.kandimalla@iict.res.in; 17Department of Biochemistry, Kakatiya Medical College, Warangal, Telangana 500022, India; 18School of Pharmacy and Pharmaceutical Science, Ulster University, Coleraine BT52 1SA, Northern Ireland, UK; m.tambuwala@ulster.ac.uk; 19Department of Zoology, Patna University, Patna, Bihar 800005, India; gkazad@patnauniversity.ac.in; 20Department of Environmental Health Sciences, Tulane University, New Orleans, LA 70112, USA; sshercha@tulane.edu; 21Translational Laboratory in Molecular Physiology, Centre for Experimental Surgery, College of Medicine, Federal University of Rio de Janeiro (UFRJ), Rio de Janeiro 21941901, Brazil; wagner.baetas@gmail.com; 22Center for iPS Cell Research and Application, Kyoto University, Kyoto 606-8501, Japan; kazuo.takayama@cira.kyoto-u.ac.jp; 23Biomaterials and Bioengineering Lab, Translational Research Centre San Alberto Magno, Catholic University of Valencia San Vicente Mártir, c/Guillem de Castro 94, 46001 Valencia, Spain; angel.serrano@ucv.es; 24School of Engineering and Sciences, Tecnologico de Monterrey, Av. Eugenio Garza Sada 2501 Sur, Monterrey 64849, Nuevo León, Mexico; gchauhan@tec.mx; 25Department of Molecular Medicine, University of Padova, Via Gabelli 63, 35121 Padova, Italy; giorgio.palu@unipd.it; 26Division of Hematology/Oncology, Department of Medicine, UPMC Hillman Cancer Center, University of Pittsburgh School of Medicine, Pittsburgh, PA 15260, USA; brufskyam@upmc.edu

**Keywords:** ACE2, viral spike receptor-binding domain, SARS-CoV-2, transmission, bioinformatics

## Abstract

Angiotensin-converting enzyme 2 (ACE2) is the cellular receptor for the Severe Acute Respiratory Syndrome Coronavirus 2 (SARS-CoV-2) that is engendering the severe coronavirus disease 2019 (COVID-19) pandemic. The spike (S) protein receptor-binding domain (RBD) of SARS-CoV-2 binds to the three sub-domains viz. amino acids (aa) 22–42, aa 79–84, and aa 330–393 of ACE2 on human cells to initiate entry. It was reported earlier that the receptor utilization capacity of ACE2 proteins from different species, such as cats, chimpanzees, dogs, and cattle, are different. A comprehensive analysis of ACE2 receptors of nineteen species was carried out in this study, and the findings propose a possible SARS-CoV-2 transmission flow across these nineteen species.

## 1. Introduction

We had been acquainted with the term beta-coronavirus for about two decades when we first encountered the Severe Acute Respiratory Syndrome Coronavirus (SARS-CoV) outbreak that emerged in 2002, infecting about 8000 people with a 10% mortality rate [1]. It was followed by the emergence of the Middle East Respiratory Syndrome Coronavirus (MERS-CoV) in 2012 with 2300 cases and mortality rate of 35% [2]. The third outbreak, caused by SARS-CoV-2, was first reported in December 2019 in China, Wuhan province, which rapidly took the form of a pandemic [3,4,5]. To date, this new human coronavirus has affected 65.5 million people worldwide and is held accountable for over 1.5 million deaths [6]. SARS-CoV-2 is an enveloped single-stranded plus sense RNA virus whose genome is about 30 kb in length, and which encodes for 16 non-structural proteins, four structural, and six accessory proteins [7]. The four major structural proteins which play a vital role in viral pathogenesis are Spike protein (S), Nucleocapsid protein (N), Membrane protein (M), and Envelope protein (E) [8,9]. SARS-CoV-2 infection is mainly characterized by pneumonia [10]; however, multi-organ failure involving myocardial infarction, hepatic, and renal damage is also reported in patients infected with this virus [11]. SARS-CoV-2 binds to the Angiotensin-converting enzyme 2 (ACE2) receptor on the host cell surface via its S protein [12,13]. ACE2 plays an essential role in viral attachment and entry [14,15]. The study of the interaction of ACE2 and S protein is of utmost importance [16,17,18].

The S1 subunit of the S protein has two domains, the C-terminal and the N-terminal domains, which fold independently, and either of the domains can act as Receptor Binding Domain (RBD) for the interaction and binding to the ACE2 receptor widely expressed on the surface of many cell types of the host [19,20]. The human ACE2 protein is 805 amino acids long, containing two functional domains: the extracellular N-terminal claw-like peptidase M2 domain and the C-terminal transmembrane collectrin domain with a cytosolic tail [21]. The RBD of the S protein binds to three different regions of ACE2, which are located at amino acids (aa) 24–42, 79–84, and 330–393 positions present in the claw-like peptidase domain of ACE2 [14]. These binding regions are designated in our study as domains D1, D2, and D3, respectively. ACE2 modulates angiotensin activities, which promote aldosterone release and increase blood pressure and inflammation, thus causing damage to blood vessel linings and various types of tissue injury [22]. ACE2 converts Angiotensin II to other molecules and reduces this effect [23]. However, when SARS-CoV-2 binds to ACE2, the function of ACE2 is inhibited and, in turn, leads to endocytosis of the virus particle into the host cell [24].

Zoonotic transmission of this virus from bat to human and random mutations acquired by SARS-CoV-2 during human to human transmission has also empowered this virus with the ability to undergo interspecies transmission, and, recently, many cases have been reported stating that different species can be infected by this virus [25,26].

In this study, we aim to determine the susceptibility of other species, whether they bear the capability of being a possible host of SARS-CoV-2. We chose nineteen different species (*Bos taurus*, *Capra hircus*, *Danio rerio*, *Equus caballus*, *Felis catus*, *Gallus gallus*, *Homo sapiens*, *Macaca mulatta*, *Manis javanica*, *Mesocricetus auratus*, *Mustela putorius furo*, *Pelodiscus sinensis*, *Pteropus alecto*, *Pteropus vampyrus*, *Pan troglodytes*, *Rattus norvegicus*, *Rhinolophus ferrumequinum*, *Salmo salar*, and *Sus scrofa*) and analyzed the ACE2 protein sequence from eighteen non-human species in relation to the human ACE2 sequence and determined the degree of variability by which the sequences differed from each other. We performed a comprehensive bioinformatics analysis in addition to the phylogenetic analysis based on full-length sequence homology, polarity along with individual domain sequence homology and secondary structure prediction of these protein sequences. These findings could have emerged to six distinct clusters of nineteen species based on the collective analysis and thereby provided a prediction of the interspecies SARS-CoV-2 transmission.

## 2. Results

Based on amino acid homology, secondary structures, bioinformatics, and polarity of the three domains D1, D2, and D3 of ACE2, all nineteen species were clustered. Note that, since ACE2 from Salmo salar is missing 119 N-terminal residues, this protein was not included in the analysis of the sequence conservation of the D1 (residues 24–42) and D2 domains (residues 79–84) involved in the interaction with SARS-CoV spike glycoprotein. Finally, a cumulative set of nineteen species clusters was built, among which the SARS-CoV-2 transmission may occur.

### 2.1. Phylogeny and Clustering Based on ACE2 Domain-Based Homology

First, we examined all the substitutions with similar properties and similar side chain binding atoms, signifying that the substitutions would not impede the SARS-CoV-2 transmission. Note that all the mutations are considered concerning the human ACE2 domains D1, D2, and D3 (Figure 1).

In D1 domain: out of eighteen species, eight species were found to possess a substitution at position 30 where D (aspartate) was substituted by E (glutamate), and four species were found to carry the D38E substitution. It was reported that, in the aspartate side chain, the oxygen atom was involved in ionic-ionic interaction and the side-chain oxygen atom was also present in glutamate, so this substitution may not affect the protein–protein interaction properties [27,28]. In the T27S substitution, threonine and serine both possess OH that participates in binding, and in the H34L substitution, both histidine and leucine use the NH group for interaction with another amino acid (backbone HN). Consequently, if we consider only the critical perspective for these substitutions, we can conclude that these changes would not impede the binding between the S and ACE2 protein.

In D2 domain: L79I bears importance across eighteen species since both of these amino acids (leucine and isoleucine) share similar chemical properties. Thus, if we analyze the changes in amino acid residues based on their chemical properties, which is the main contributing factor for protein–protein interaction, we can conclude that it will not significantly affect the binding between ACE-2 and RBD of the S protein.

In D3 domain: out of eleven substitutions, three substitutions (R393K, K353H, and K353R) were observed of the similar type with similar side chain interacting atoms and therefore changed at these positions would not affect the interaction of ACE2 with that of the S protein.

Secondly, across all nineteen species, homology was derived based on amino acid sequences, and, consequently, associated phylogenetic trees were drawn (Figure 2).

Six clusters of the nineteen species were formed using the K-means clustering technique based on sequence homology of the three domains (Figure 3). The clusters of species {S1,S2,S3} and {S6,S13} stayed together for the ACE2 full-length sequence homology and the combination of three domain-based sequence similarity. The species S16, S17, and S18 also followed the same as observed.

Furthermore, it was observed that sequence homology of the D1, D2, and D3 domains clustered the species S15 into the cluster where S9, S10, and S12 belong, although S15 was similar ACE2 sequence of S8 and S9. Despite S4 being very similar to S9, S10, and S12 for full-length ACE2 homology, it combined with S5 and S11 concerning the three domain-based sequence spatial organizations. In addition, S7 was found to be in the proximity of S6 and S13 although S7 was very much similar to S5 and S11 based on ACE2 homology.

### 2.2. Clustering Based on Secondary Structures

For each existing domain of ACE2 of the nineteen species, the secondary structure was predicted (Figure 4). For each domain, species are grouped into several subgroups.

Concerning the D1 domain:*Bos taurus* and *Capra hircus**Equus caballus* and *Felis catus**Mustela putorius furo* has a structure closer to *Equus caballus* and *Felis catus* as it has only one difference of a coil present at position 42. In addition, *Mesocricetus auratus* has a secondary structure similar to the above two, except an extended helix at position four. Similarly, *Sus scrofa* has an extended sheet instead of a helix at position 42. Thus, *Mustela putorius furo*, *Mesocricetus auratus*, and *Sus scrofa* can be put in the same cluster as *Equus caballus* and *Felis catus*.*Homo sapiens*, *Macaca mulatta*, and *Pan troglodytes**Manis javanica* and *Rhinolophus ferrumequinum**Pteropus alecto* and *Pteropus Vampyrus**Rattus norvegicus* and *Pelodiscus sinensis* have similar structures differing by the presence of an extra coil at position 39 for *Rattus norvegicus*.*Gallus gallus* and *Danio rerio* have a unique secondary structure in comparison to the others.

These individual eight clusters show six different secondary structures in D1 shared by sixteen species, which shows high similarities in their secondary structures, while the remaining two have a unique secondary structure for D1 domain. Thus, these eight clusters have similar secondary structures indicating that the species in the eight clusters are closely related.

With respect to the D2 domain:*Homo sapiens*, *Macaca mulatta*, and *Pan troglodytes**Bos taurus*, *Mustela putorius furo*, *Pteropus alecto*, and *Pteropus vampyrus**Equus caballus*, *Felis catus*, *Manis javanica*, *Pelodiscus sinensis*, and *Rhinolophus ferrumequinum**Danio rerio* and *Gallus gallus*

Similarly, for the D2 domain, we found four clusters with the same secondary structure, indicating that they are closely related.

With respect to the D3 domain:*Homo sapiens* and *Pan troglodytes**Bos Taurus*, *Rhinolophus ferrumequinum*, *Sus scrofa*, and *Capra hircus**Equus caballus* and *Felis catus**Pteropus alecto* and *Pteropus vampyrus*

Again for the D3 domain, four different clusters were bearing similar secondary structures; therefore, these species are also closely related.

Based on the similarity among the three domains, all eighteen species were clustered (Figure 5).

From the clusters (Figure 5) based on the secondary structure of the three domains of ACE2, it was observed that the species S4 was clustered uniquely, though S4 is clustered with S9 and S19 based on ACE2 full-length sequence homology. Furthermore, S6 and S13 were found to be similar based on ACE2 homology, but they got clustered into two different clusters when the secondary structure of three domains was concerned. In contrast, the group of species {S1,S2,S3}, {S9,S12}, {S16,S18}, and {S5,S7,S11} remained in the same clusters concerning ACE2 homology as well as individual secondary structures of the domains.

### 2.3. Clustering Based on Bioinformatics

Twelve bioinformatics features viz. Shannon entropy, instability index, aliphatic index, charged residues, half-life, melting temperature, N-terminal of the sequence, molecular weight, extinction coefficient, net charge at pH7, and isoelectric point of the D1, D2, and D3 domains of ACE2 for all nineteen species were determined (Figure 6).

For each species, a twelve-dimensional feature vector was found (Figure 6). For each domain D1, D2, and D3 domain, a distance matrix was determined using the Euclidean distance
$$d\left(S,T\right)=\sqrt{\sum_{i=1}^{12}{\left(f_{i}-g_{i}\right)}^{2}}$$Note that here $f_{i}$ and $g_{i}$ denote the *i*th feature for the species *S* and *T*, respectively. These distance matrices with heatmap representation for all three domains are presented in Figure 7, Figure 8 and Figure 9. In addition, by inputting the distance matrix, using the K-means clustering technique, several clusters of species were formed for D1 and D2 domains in eighteen species (Figure 7 and Figure 8) and D3 domain in all nineteen species (Figure 9).

A final set of six clusters was formed using the K-means clustering method to have all three domains for eighteen different species (Figure 10). Although the species S7 was clustered with the species S5 and S11 as per full-length ACE2 sequence homology, S7 formed a unique singleton cluster when the bioinformatics features were taken into consideration. Similarly, the species S16 formed a singleton cluster though it was clustered with S17, S18, and S19 as per the amino acid homology of ACE2. The sequence homology of ACE2 made the four species S16, S17, S18, and S19 into a single cluster, but bioinformatics features placed the species S18 in a cluster where the other three species S1, S2, and S3 belonged. Based on bioinformatics features, S4 clustered together with S15 though the ACE2 receptor of S4 was sequentially similar to ACE2 of S9, S10, and S12.

The clusters {S1,S2,S3}, {S6,S13}, and {S9,S10,S12} were unaltered with respect to the full length ACE2 homology and bioinformatics features.

### 2.4. Phylogeny and Clustering Based on Polarity

In the D1 domain, it was observed that the polarity of thirteen amino acids among nineteen (24–42 aa) amino acids were found to be conserved across eighteen species. Based on the amino acids’ polarity and non-polarity nature, the species were arranged in a phylogenetic tree (Figure 11).

It was found that *Homo sapiens*, *Pan troglodyte*, *Macaca mulatta*, and *Danio rerio* closer according to this analysis. *Pteropus alecto*, *Pteropus vampyrus*, and *Sus scrofa* occurred in parallel along with the above three and formed a different clade indicating the closeness based on polarity. Again, the case for *Gallus gallus*, *Rhinolophus ferrumequinum*, *Mustela putorius furo*, *Equus caballus*, and *Felis catus* is similar. Two separate groups, *Mesocricetus auratus*, *Manis javanica*, and *Capra hircus*, *Bos taurus*, were similarly placed nearby, indicating that the polarity of amino acids of the proteins for these species was similar. *Pelodiscus sinensis* and *Rattus norvegicus* occurred separately and were not grouped with any other species but bears similarity with both the groups containing species *Mesocricetus auratus*, *Manis javanica*, *Pteropus alecto*, *Pteropus vampyrus*, and *Sus scrofa*.

In the D2 domain, out of six amino acid long sequences, the polarity of three amino acids was conserved across eighteen species, and among them, one amino acid was a binding residue. *Homo sapiens*, *Pan troglodytes*, *Macaca mulatta*, *Pteropus vampyrus*, and *Pteropus alecto* were grouped together since the overall polarity of their amino acid chain was found to be similar, and, simultaneously, *Danio rerio*, *Mustela putorius furo*, *Gallus gallus*, and *Pelodiscus sinensis* were placed together. In addition, three groups comprising *Manis javanica*, *Capra hircus*, *Bos taurus*, *Sus scrofa*, *Rattus norvegicus*, and *Rhinolophus ferrumequinum*, respectively, were placed in close proximity based on their polarity and non-polarity of the amino acids in the protein sequence. However, *Equus caballus*, *Felis cattus*, and *Mesocricetus auratus* were placed separately since they did not show much resemblance based on polarity.

In the D3 domain of ACE2 sequences of *Salmo salar* and *Danio rerio*, there was an insertion of a polar amino acid into one of the binding residue positions that may affect the binding of ACE2 to that of RBD of SARS-CoV2 negatively. A total of three binding residues were already reported in the D3 domain, of which one of them remained conserved concerning polarity across the nineteen species. *Rattus norvegicus, Mustela putorius furo, Mesocricetus auratus*, and *Felis catus* were grouped under a single clade based on the polarity of their protein sequence. It was a similar case for *Danio rerio*, *Pelodiscus sinensis*, *Salmo salar*, and *Gallus gallus*. Due to the sequence similarity between *Pteropus vampyrus* and *Pteropus alecto*, their polarity of the protein sequence was also similar and thus grouped. Sequence similarity was also observed for *Homo sapiens*, *Pan troglodytes*, and *Macaca mulatta*, so again these were categorized together. Two groups comprised of *Rhinolophus ferrumequinum*, *Capra hircus*, and *Bos taurus*, *Sus scrofa*, respectively, were sorted together indicating their similar nature of polarity and non-polarity of protein sequence. Lastly, *Manis javanica* and *Equus caballus* were placed separately signifying that the sequences of both species were quite distinct.

The individual groups of species based on the polarity of individual D1, D2, and D3 domains have emerged into six disjoint clusters (Figure 12).

Here, the clusters {S1,S2,S3}, {S16,S17,S18}, {S8,S14}, {S6,S13}, and {S10,S12} remained invariant with regard to the homology of full length ACE2 as well as polarity sequence of the D1, D2, and D3 domains.

### 2.5. Possible Clusters of Transmission of SARS-CoV-2

Based on all the different clusters formed on the basis of amino acid homology, secondary structures, bioinformatics, and polarity of the D1, D2, and D3 domains of ACE2, final clusters of all nineteen species were devised using the K-means clustering method Figure 13.

In Figure 13, it was found that the cluster-1 (C-1) comprising of *Homo sapiens*, *Pan troglodyte*, and *Macaca mulatta* were close to cluster-5 (C-5) comprising of *Felis catus* (Cat), *Mesocricetus auratus* (Golden Hamster), *Manis javanica* (Sunda pangolin), *Mustela putorius furo* (Ferret), *Rattus norvegicus* (Rat), and *Rhinolphus ferrumequinum* (Greater horseshoe bat) (Figure 13). This C-5 is also close to cluster-3 (C-3) [*Gallus gallus* (red jungle fowl), *Pelodiscus sinensis* (Chinese shell turtle), *Danio rerio* (zebrafish) and *Salmo salar*], and cluster-4 (C-4) [*Capra hircus* (Goat), *Bos taurus* (Cattle), and *Sus scrofa* (pig)]. C-4 also showed resemblance with cluster-2 (C-2) [*Pteropus alecto*, *Pteropus vampyrus*], and cluster-6 (C-6) that is comprised of *Equus caballus* (horse) only. However, both C-2 and C-6 were also close to each other.

Furthermore, pooled analyses based on the two types of substitutions (one is affecting SARS-CoV-2 transmission (M1), and the other one is SARS-CoV-2 non-affecting transmission(M2)) for all six of the final clusters, which are presented in Table 1.

Based on Table 1, information regarding the number of M1 and M2 substitutions and the intra-species transmission of SARS-CoV-2 were presented as follows:C-1: None of the species bear any mutation in the binding residues and are conserved, so viral transmission is immaculate.C-2: This cluster has an equal number of transmission affecting and transmission non-affecting types of substitutions. Therefore, both have an equal probability of getting infected from each other.C-3: Here, again, *Gallus gallus*, *Pelodiscus sinensis*, and *Danio rerio* have a similar ratio of S1 to S2, signifying possible flow of viral transmission within these three species. However, *Salmo salar* is unique and distant, and therefore, the probability of viral transmission is unlikely.C-4: The species in this cluster have a similar number of transmission-affecting and transmission non-affecting types of substitutions show that the flow of viral transmission would be continuous among these three species.C-5: Transmission between *Felis catus* and *Mesocrietus auratus* is highly likely, which is the same for *Manis javanica*, *Mustela putorius furo*, and *Rattus norvegicus* as indicated by their similar number of substitutions. Therefore, the inter-transmission between these species is highly plausible. While *Rhinolophus ferrumequinum* has a relatively high value of transmission affecting substitutions from all of the above, its susceptibility to getting infected from other species is uncertain.C-6: A total of five transmission affecting substitutions in the three domains for *Homo sapiens* were observed.

## 3. Discussion

In this study, we amassed the ACE2 protein sequences of nineteen species to investigate the possible transmission of SARS-CoV-2 among these species in relation to human ACE2 protein. Multiple sequence alignments of these ACE2 receptors enabled us to estimate the similarity concerning amino acids and, from that, we observed that *Salmo salar* (Salmon fish) was quite distant. It also gave us the idea that some of the amino acid substitutions in the binding residues occurring across the species with respect to human ACE2 resulted in amino acids have similar binding properties, indicating that their interactions with RBD of the S protein will be similar to that of humans, thus making transmission across these species feasible. It was observed that ACE2 sequences from *Homo sapiens* and *Pan troglodytes* (Chimpanzee) were almost identical (showing 99.01% sequence identity). Although ACE2 from *Macaca mulatta* (Rhesus macaque) also shared a high percentage of sequence identity with human protein (95.16%), it possesses substitutions at 39 positions. However, no substitutions were observed in the amino acid residues involved in the interaction with the RBD of the S protein, making the viral transmission across these species highly likely. Again, *Pteropus vampyrus* (Large flying fox) and *Pteropus alecto* (Black flying fox) have precisely the same ACE2 sequence, and thus signifying high viral transmission and that both of them have an equal chance of getting infected by each other.

Further analysis led us to present a possible transmission flow among the nineteen species, as illustrated in Figure 13. The multifaceted examination of the ACE2 protein indicated that interspecies SARS-CoV-2 transmission is quite possible, and we have tried to provide a better insight into it by predicting the possible transmission among species within the same cluster and between clusters too. However, further in-depth analysis is necessary in the future for the identification of new hosts of SARS-CoV-2 as well as for determination of possible ways to prevent inter-species transmission.

The results reported in this study allow us to propose possible routes of the SARS-CoV-2 transmission flow among species. Unsurprisingly, our results indicate that, among the species studied, it is the members of primates that are the most at risk, followed by those of carnivores, cetartiodactyls, and finally bats. It is settling to see that the predicted transmission flow based on the results of our analyses is in line with the conventional evolutionary knowledge and reported infection cases. One should keep in mind though that the major goal of this study was to provide formally comprehensive structural evidence that could help in clarifying why some hosts are more susceptible than others to SARS-CoV-2 and could constitute a reservoir for further virus spillover. Obviously, more detailed studies are needed in the future to take into account structural properties of ACE2 and peculiarities of its interaction with the RBD of the S protein [15,16,17,18], and the presence of different ACE2 isoforms in individual animal species (e.g., humans have at least five ACE2 isoforms [26]). Moreover, one should consider the epigenetic regulation and expression determination of ACE2 (e.g., despite having the same protein sequence, ACE2 is differently expressed in different human cells, and different levels of expression of ACE2 are found in the same type of nasal epithelial cells or pneumocytes from humans and mice). It will also be necessary to analyze more ACE2 sequences from other species and to investigate the possibility for these different species of transforming themselves, in the long term, into healthy carriers of the virus or even into transmitters and diffusers of the disease.

It is well known that protein pairs with a sequence identity greater than 40% are very likely to be structurally similar, whereas protein pairs with a sequence identity of 20–35% represent a ‘twilight zone’, where structural similarity in pairs is considerably less common, with less than 10% of protein pairs with sequence identity below 25% have similar structures [29,30,31]. Sequence identity of the ACE2 proteins from nineteen species analyzed in this study ranges from 99.01% (*Homo sapiens* vs. *Pan troglodytes*) to 58.0% (*Homo sapiens* vs. *Danio rerio*), with the lowest identity of 57.13% being between the proteins from *Danio rerio* and *Rhinolophus ferrumequinum*. Therefore, one might expect rather close overall structural organization of all these proteins, even the most distant ones. In fact, even the lowest level of sequence identity for the pair of ACE2 proteins is still well above the sequence identity of 20–35% characteristics for the ‘twilight zone’. On the other hand, fold-level, global structural similarity does not exclude the presence of local structural variability that might define, for example, the peculiarities of protein–protein interactions. Structural information is currently available only for the ACE2 from *Homo sapiens* and *Felis catus*. Therefore, previous studies that analyzed the peculiarities of interactions between the viral spike protein and host ACE2 from many household and other animals, such as *Pan troglodytes* (chimpanzee), *Macaca mulatta* (Rhesus monkey), *Felis catus* (domestic cat), *Equus caballus* (horse), *Oryctolagus cuniculus* (rabbit), *Canis lupus familiaris* (dog), *Sus scrofa* (pig), *Avis aries* (sheep), *Bos taurus* (cattle), *Mus musculus* (house mouse), and *Mustela putorius furo* (ferret) [32,33,34] were focused on the structural part of these interactions and utilized a typical set of structural biology approaches, such as homology modelling and docking. Therefore, in line with our previous study [25], we decided to compare the peculiarities of the per-residue intrinsic disorder predispositions of the ACE2 proteins from nineteen species analyzed in this study rather than building their homology models. Figure 14 summarizes the results of this analysis and shows that, although these proteins have rather similar intrinsic disorder predispositions, their disorder profiles are not identical.

Furthermore, such differences in the intrinsic disorder predisposition are not equally spread through the protein sequences, with some regions (e.g., the N-terminal 150 residues and residues 500–700) of the disorder profiles showing rather noticeable variability. Figure 14 also shows that the S protein binding domains D1 and D2 of ACE2 proteins are characterized by high variability of their intrinsic disorder predispositions, whereas D3 domains are more conserved. We also looked at the peculiarities of intrinsic disorder profiles of ACE2 proteins in six clusters with the major focus at the S protein binding domains D1, D2, and D3 (see Figure 15).

This comparison revealed that the in-cluster variability of intrinsic disorder propensity was noticeably lower than the diversity between the clusters as a rule. These observations support the notion that the capability of ACE2 to interact with SARS-CoV-2 protein S can be dependent on the peculiarities of the ACE2 local intrinsic disorder predisposition [25].

## 4. Materials and Methods

### 4.1. Data Acquisition and Findings

The ACE2 protein receptor sequences from nineteen species *Homo sapiens* (Human), *Capra hircus* (Domestic goat), *Pan troglodytes* (Chimpanzee), *Equus caballus* (Horse), *Salmo salar* (Atlantic salmon), *Mesocricetus auratus* (Golden hamster), *Rhinolophus ferrumequinum* (Greater horseshoe bat), *Pteropus alecto* (Black flying fox), *Mustela putorius furo* (Domestic ferret), *Danio rerio* (Zebrafish), *Manis javanica* (Sunda pangolin), *Sus scrofa* (Domestic pig), *Macaca mulatta* (Rhesus macaque), *Bos taurus* (Aurochs), *Pelodiscus sinensis* (Chinese soft-shelled turtle), *Pteropus vampyrus* (Large flying fox), *Rattus norvegicus* (Brown rat), *Felis catus* (Domestic cat), and *Gallus gallus* (Red jungle fowl) were derived from the NCBI database [35]. Nineteen species and their respective ACE2 protein accession IDs with length are presented in Table 2.

The nearest neighborhood phylogeny of the nineteen species derived from the NCBI public server based on ACE2 protein sequence similarity is shown in Figure 16A [36].

ACE2 sequence similarity among the species derives six clusters as shown in (Figure 16B). The contact residues of the receptor-binding domain (RBD) of the spike protein (YP_009724390.1) of SARS-CoV-2 with the homo sapiens ACE2 interface are presented in Table 3 [14].

The three designated domains, D1 (aa 24–42), D2 (aa 79–84), and D3 (aa 330–393) respectively contain the residues which bind to the RBD of the S protein.

### 4.2. Methods

Examining amino acid substitutions: For human ACE2 receptor, substitutions were examined for all species, and only those substitutions are accounted for, which occurred in the binding residues in the mentioned three domains D1, D2, and D3 [14]. Based on the character of the substitutions which interfered with the binding residues of the ACE2 across various species, two types were defined: substitutions affected transmission (M1) and substitutions which did not affect transmission (M2).

Multiple sequence alignments and associated phylogenetic trees were developed using the NCBI web-suite across all individual binding domains D1, D2, and D3 in eighteen species and D3 in *Salmo salar* [37,38].

K-means clustering: The algorithmic clustering technique derives homogeneous subclasses within the data such that data points in each cluster are as similar as possible according to a widely used distance measure viz. Euclidean distance. One of the most commonly used simple clustering techniques is the *K-means clustering* [39,40]. The algorithm is described below in brief:

*Algorithm*: K-means algorithm is an iterative algorithm that tries to form equivalence classes from the feature vectors into K (pre-defined) clusters where each data point belongs to only one cluster [39].

Assign the number of desired clusters (*K*) (in the present study, K=6).Find centroids by first shuffling the dataset and then randomly selecting *K* data points for the centroids without replacement.Keep iterating until there is no change to the centroids.Find the sum of the squared distance between data points and all centroids.Assign each data point to the closest cluster (centroid).Compute the centroids for the clusters by taking the average of the all data points that belong to each cluster.

In this present study, nineteen species were clustered using *Matlab* by inputting the distance matrix derived from the feature vectors associated with the three domains of ACE2 across all species.

Secondary structure predictions: The secondary structure of full-length ACE2 sequence of all species were predicted using the web-server CFSSP (Chou and Fasman Secondary Structure Prediction Server) [41]. This server predicts secondary structure regions from the protein sequence such as alpha-helix, beta-sheet, and turns from the amino acid sequence [41]. On obtaining the full-length ACE2 secondary structures, individual domains D1, D2, and D3 were cropped for each species.

Bioinformatics features: Several bioinformatics features viz. Shannon entropy, instability index, aliphatic index, charged residues, half-life, melting temperature, N-terminal of the sequence, molecular weight, extinction coefficient, net charge at pH7, and isoelectric point of D1, D2, and D3 domains of ACE2 for all nineteen species were determined using the web-servers *Pfeature and ProtParam* [42,43].

Computational analysis of the intrinsic disorder predisposition: Per-residue propensity of the ACE2 proteins from nineteen species for the intrinsic disorder were evaluated by the PONDR^®^ VSL2 algorithm [44,45], which is one of the more accurate stand-alone per-residue disorder predictors [46,47]. In these analyses, residues with the disorder scores exceeding the threshold value of 0.5 are considered as intrinsically disordered, whereas residues with the predicted disorder scores between 0.2 and 0.5 are considered as flexible.

Shannon entropy: Shannon entropy measures the amount of complexity in a primary sequence of ACE2. It was determined using the web-server *Pfeature* by the formula
$$SE=-\sum_{i=1}^{20}p_{i}log_{2}\left(p_{i}\right),$$
where $p_{i}$ denotes the frequency probability of a given amino acid in the sequence [42].

Instability index: Instability index is determined using the web-server *ProtParam*, and it estimates the stability of a protein in a test tube. A protein whose instability index is smaller than 40 is predicted as stable. A value above 40 predicts that the protein may be unstable [42].

Aliphatic index: Aliphatic index of a protein is defined as the relative volume gathered by aliphatic side chains (alanine, valine, isoleucine, and leucine). It may be regarded as a positive factor for increasing the thermostability of globular proteins, such as ACE2 [42].

N-terminal: It was reported that the N-terminal of a protein is responsible for its function. For each domain sequence, N-terminal residue was determined using the *Pfeature* [42].

In vivo half-life: The half-life predicts the time it takes for half of the protein amount to degrade after its synthesis in the cell. The N-end rule originated from the observations that the identity of the N-terminal residue of a protein plays an essential role in determining its stability in vivo [48].

Extinction coefficients: The extinction coefficient measures how much light a protein absorbs at a particular wavelength. It is useful to estimate this coefficient when a protein is purified [48].

Polarity sequence: Every amino acid in the domains D1, D2, and D3 of ACE2 were recognized as polar (P) and non-polar (Q) and thus every D1, D2, and D3 for eighteen species and the domain of *Salmo salar* turned out to be binary sequences with two symbols P and Q. Then, homology of these sequences for each domain was made and, consequently, a phylogenetic relationship was drawn.

## Figures and Tables

**Figure 1 molecules-25-05906-f001:**
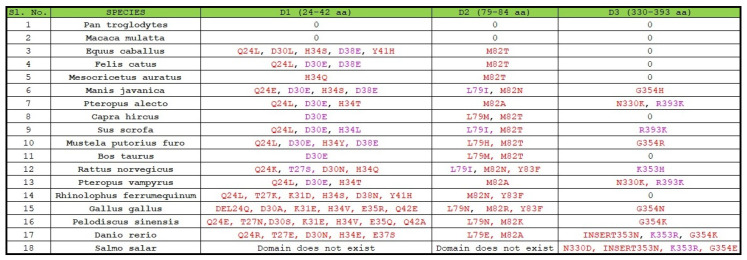
Substitutions in D1, D2, and D3 domains of ACE2 across eighteen species.

**Figure 2 molecules-25-05906-f002:**
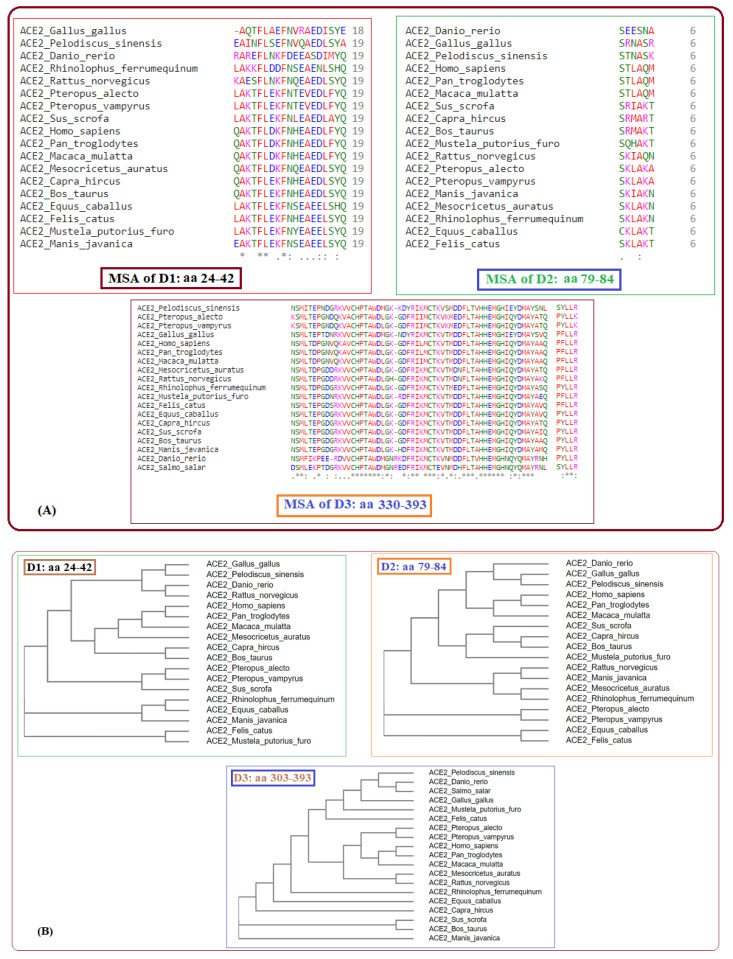
Multiple sequence alignments of D1, D2, and D3 domains of ACE2 of nineteen species (**A**) and respective phylogenies (**B**).

**Figure 3 molecules-25-05906-f003:**
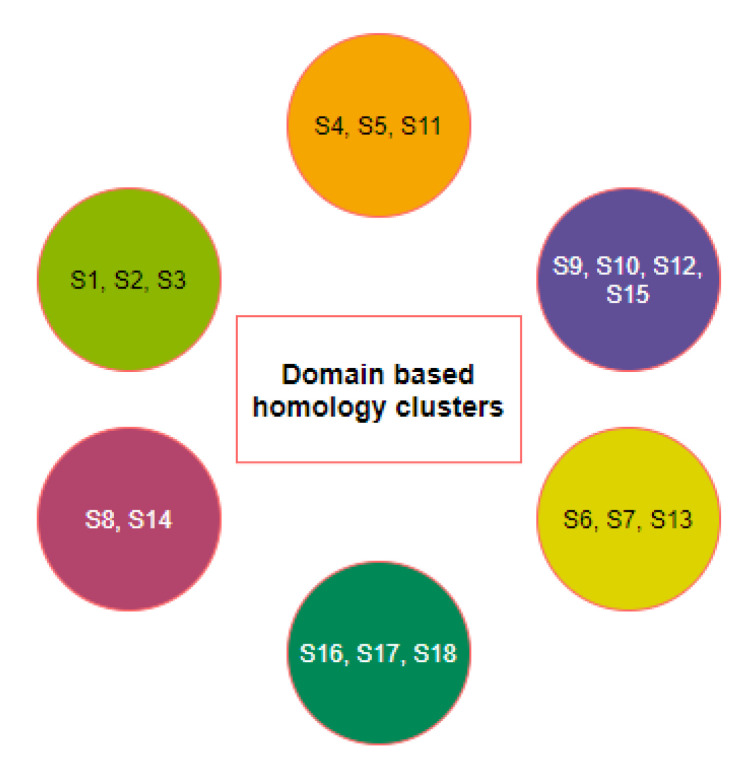
Clusters of species based on domain-based sequence homology.

**Figure 4 molecules-25-05906-f004:**
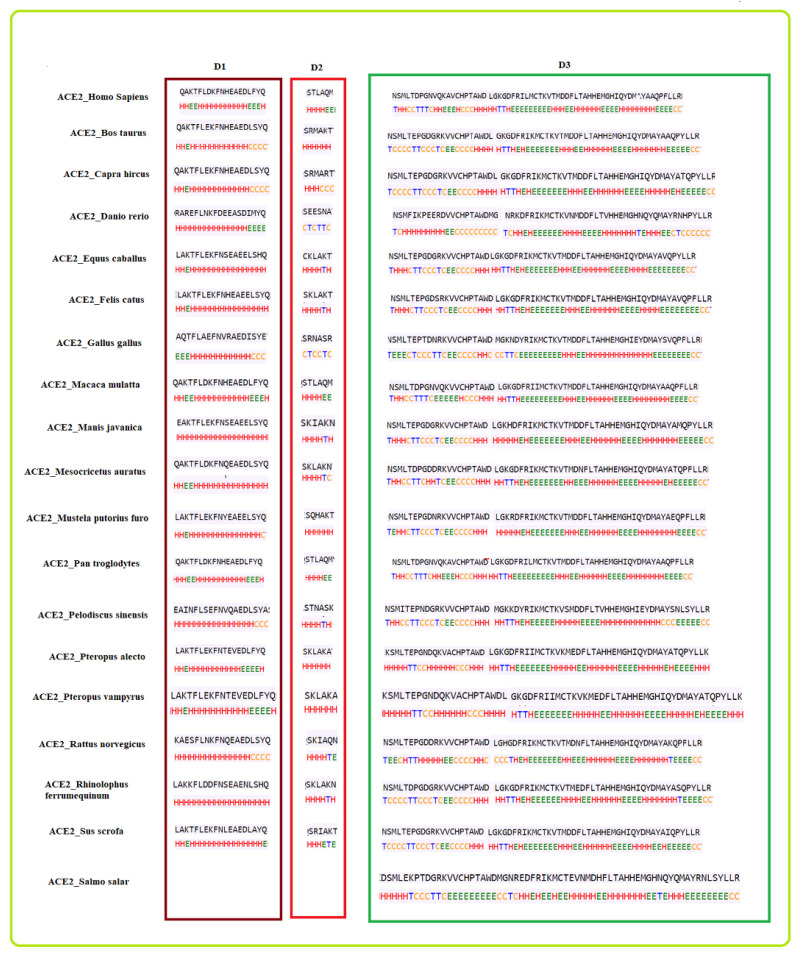
Predicted secondary structures of D1, D2, and D3 domains for 18 species and only D3 domain for *Salmo salar*.

**Figure 5 molecules-25-05906-f005:**
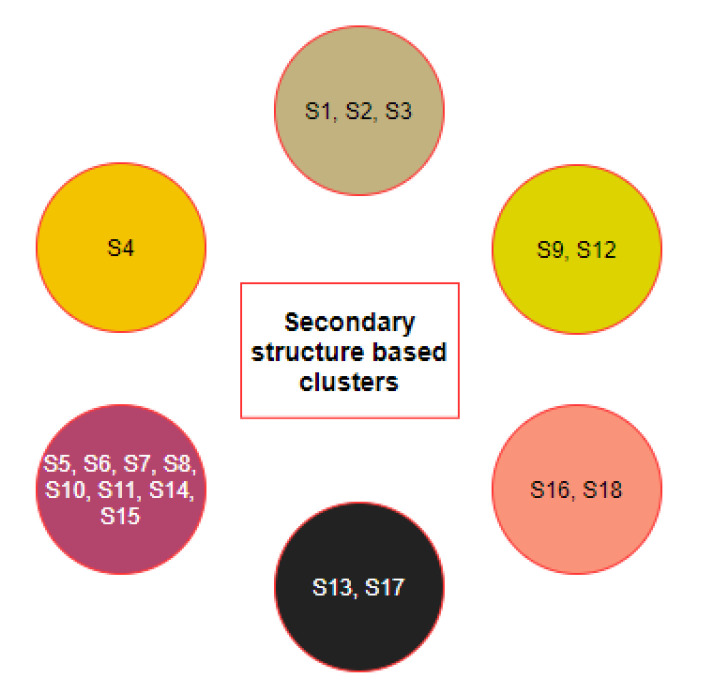
Clusters of species based on the secondary structures of the D1, D2, and D3 domains.

**Figure 6 molecules-25-05906-f006:**
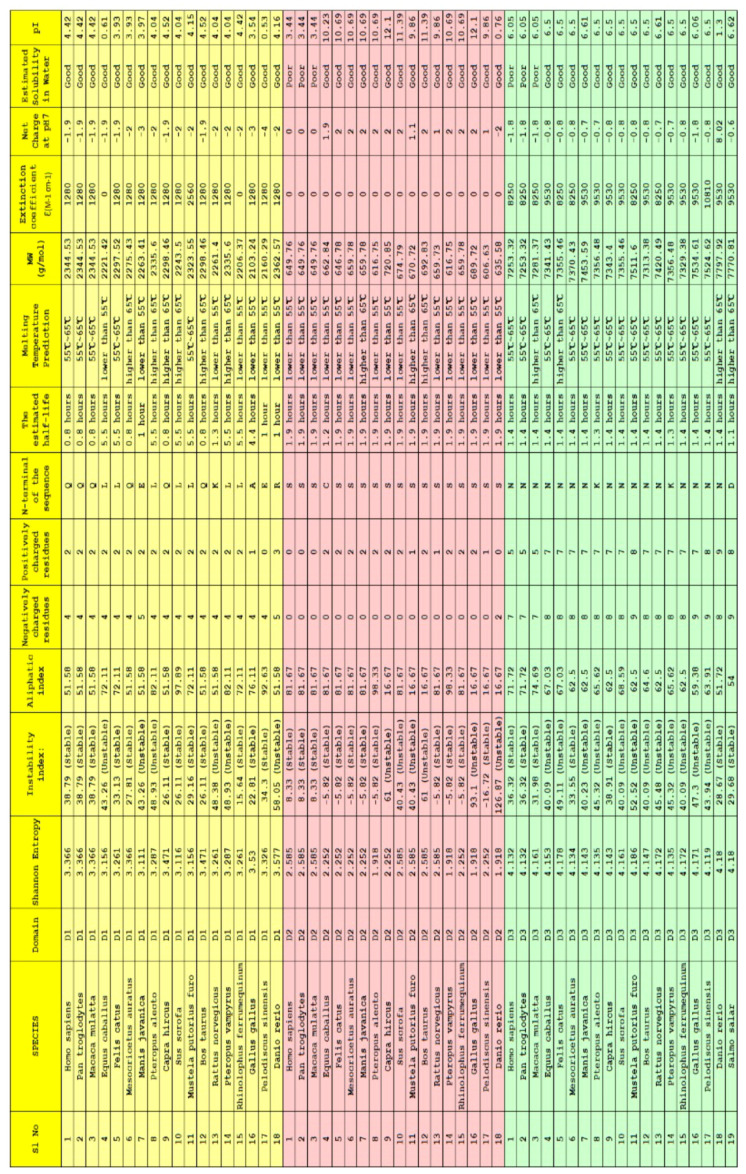
Bioinformatics of the D1, D2, and D3 domains of ACE2 from eighteen species. For *Salmo salar*, only D3 bioinformatics was presented.

**Figure 7 molecules-25-05906-f007:**
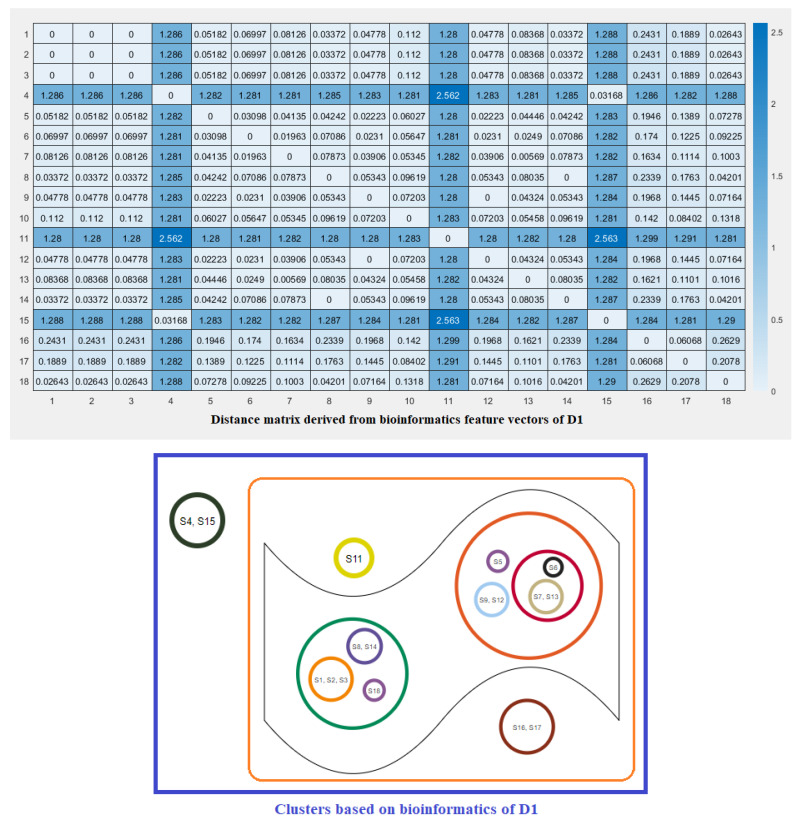
Distance matrix based on the bioinformatics feature vectors of D1 of ACE2 across eighteen eighteen species and associated clusters.

**Figure 8 molecules-25-05906-f008:**
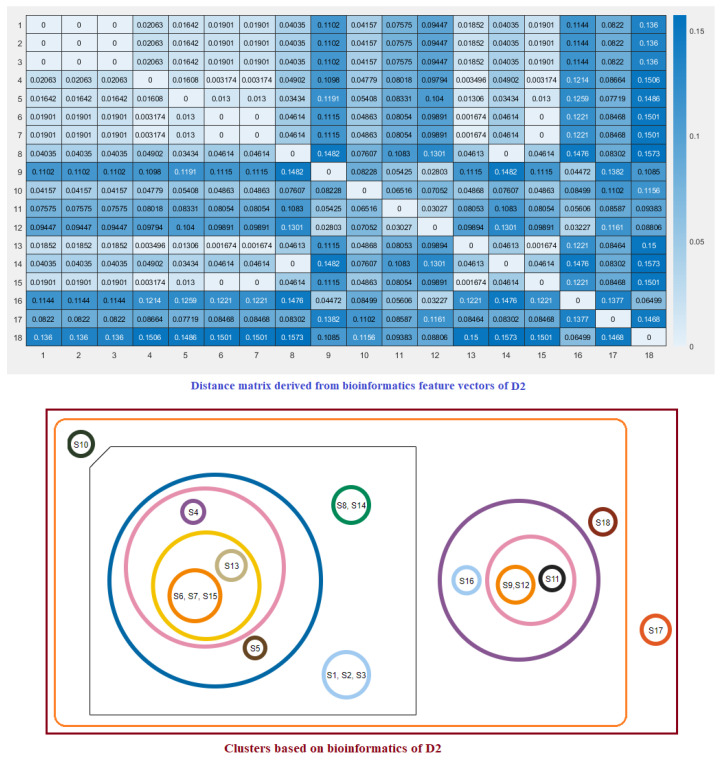
Distance matrix based on the bioinformatics feature vectors of D2 of ACE2 across eighteen species and associated clusters.

**Figure 9 molecules-25-05906-f009:**
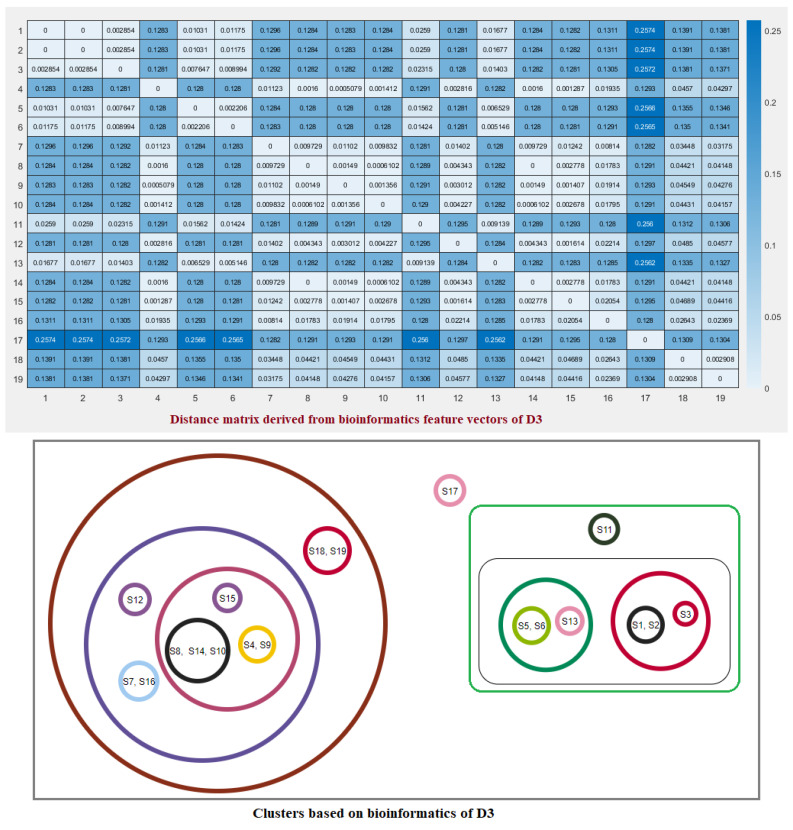
Distance matrix based on the bioinformatics feature vectors of D3 of ACE2 across nineteen species and associated clusters.

**Figure 10 molecules-25-05906-f010:**
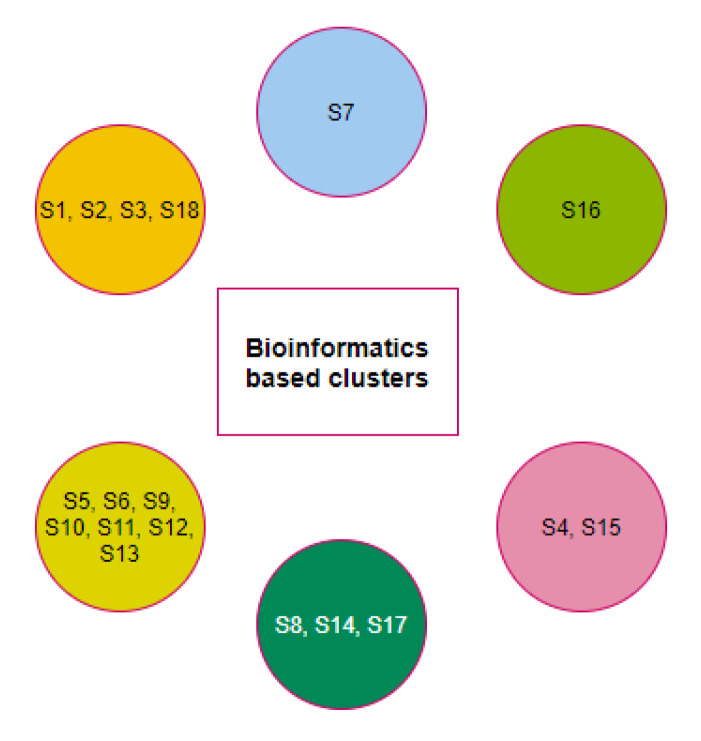
Clusters of species based on the bioinformatics of the D1, D2, and D3 domains.

**Figure 11 molecules-25-05906-f011:**
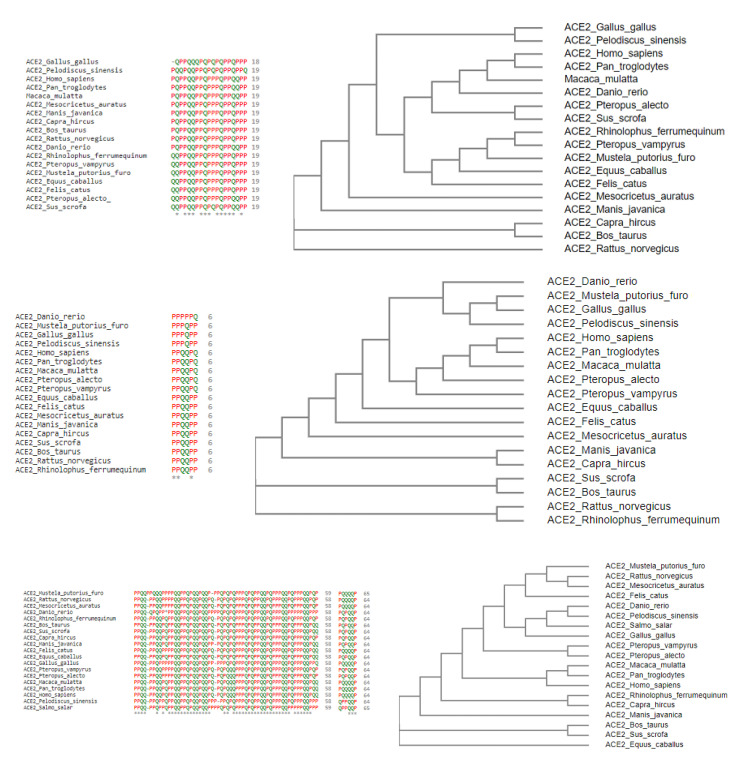
Polarity sequence of the D1, D2, and D3 domains across all species alignment and associated phylogenetic relationships.

**Figure 12 molecules-25-05906-f012:**
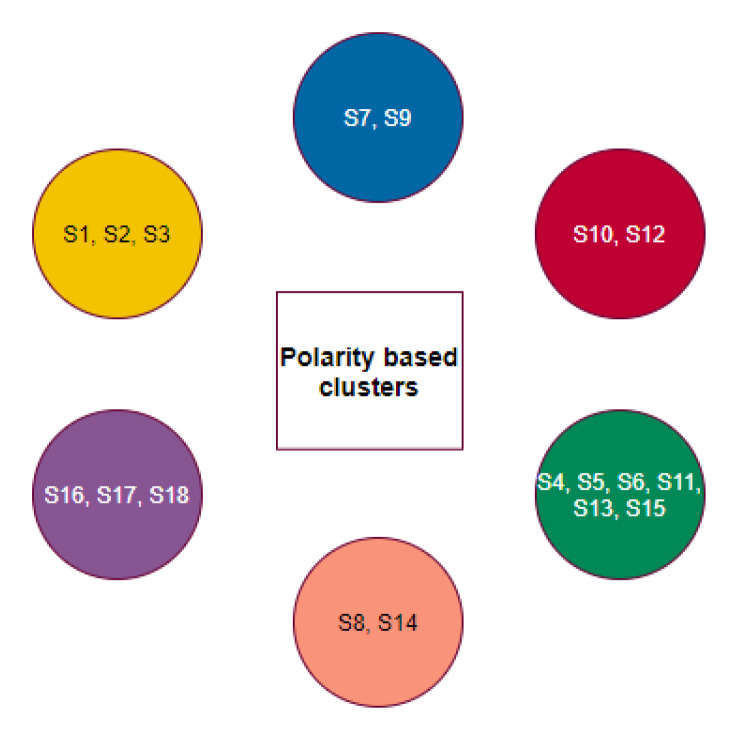
Clusters based on groups of species based on domain-wise polarity.

**Figure 13 molecules-25-05906-f013:**
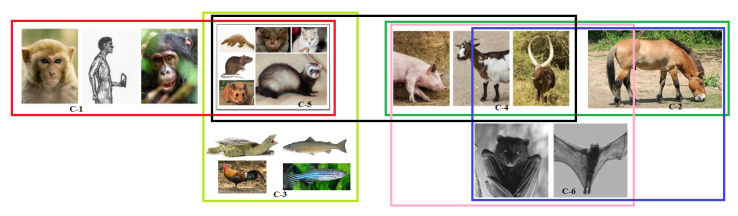
Schematic representation of a possible set of clusters of transmission of SARS-CoV-2.

**Figure 14 molecules-25-05906-f014:**
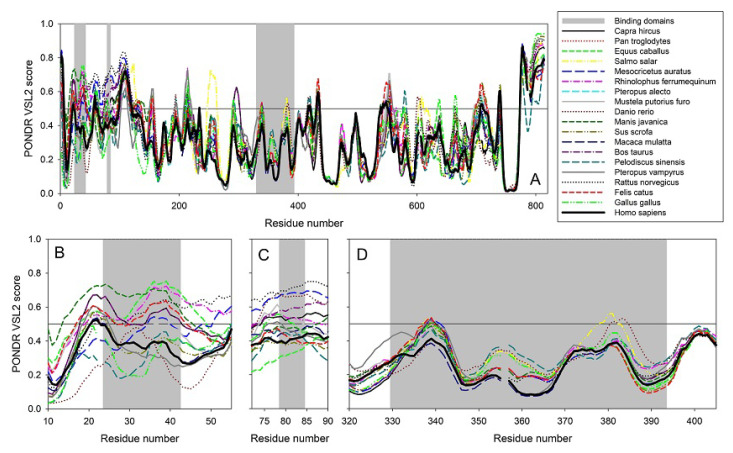
Per-residue intrinsic disorder predisposition of ACE2 proteins. (**A**) Peculiarities of the intrinsic disorder distribution within the amino sequences of ACE2 protein from nineteen species analyzed in this study. Light gray vertical bars show the location of the ACE2 regions responsible for interaction with SARS-CoV-2 S protein, domains D1 (residues 24–42), D2 (residues 79–84), and D3 (residues 330–393). (**B**–**D**). Zoomed-in disorder profiles focusing at the domains D1 (**B**), D2 (**C**), and D3 (**D**) responsible for the ACE2-S interaction. Disorder predispositions were evaluated using the PONDR^®^ VSL2 algorithm.

**Figure 15 molecules-25-05906-f015:**
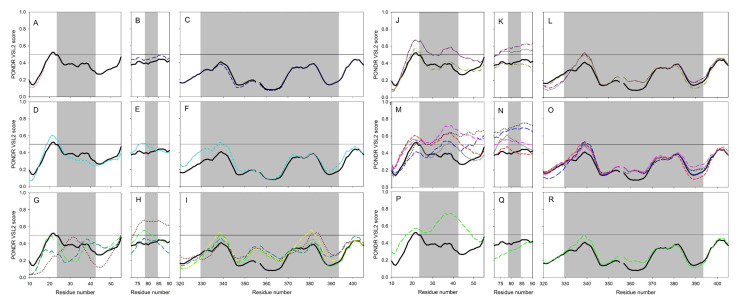
Peculiarities of intrinsic disorder predisposition within the D1 (**A**,**D**,**G**,**J**,**M**), and (**P**), D2 (**B**,**E**,**H**,**K**,**N**), and (**Q**), and D3 domains (**C**,**F**,**I**,**L**,**O**), and (**R**) of ACE2 proteins from cluster 1 (**A**,**B**), and (**C**), cluster 2 (**D**,**E**), and (**F**), cluster 3 (**G**,**H**), and (**I**), cluster 4 (**J**,**K**), and (**L**), cluster 5 (**M**,**N**), and (**O**), and cluster 6 (**P**,**Q**), and (**R**). For keys, see Figure 14.

**Figure 16 molecules-25-05906-f016:**
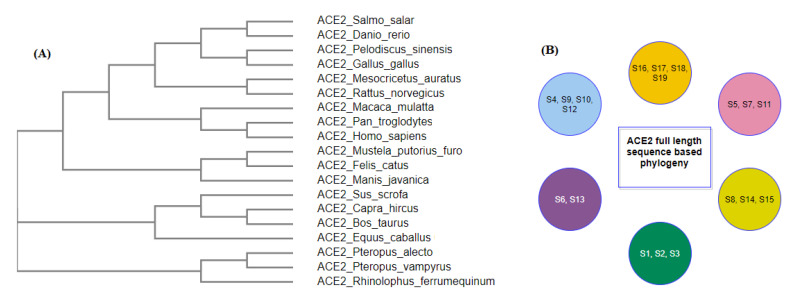
ACE2 full-length sequence-based phylogeny among nineteen species (**A**) and its derived clusters (**B**).

**Table 1 molecules-25-05906-t001:** M1 and M2 substitutions across nineteen ACE2 receptors.

Cluster	Species	DI (M1, M2)	D2 (M1, M2)	D3 (M1, M2)	Total (M1, M2)
1	*Human ACE2*	0	0	0	0
1	*Pan troglodytes*	0	0	0	0
1	*Macaca mulatta*	0	0	0	0
6	*Equus caballus*	(4,1)	(1,0)	(0,0)	(5,1)
5	*Felis catus*	(1,2)	(1,0)	(0,0)	(2,2)
5	*Mesocrietus auratus*	(1,0)	(1,0)	(0,0)	(2,0)
5	*Manis javanica*	(2,2)	(1,1)	(1,0)	(4,3)
5	*Mustela putorius furo*	(2,2)	(2,0)	(1,0)	(5,2)
5	*Rattus norvegicus*	(3,1)	(3,1)	(0,1)	(6,3)
5	*Rhinolophus ferrumequinum*	(6,0)	(2,0)	(0,0)	(8,0)
4	*Capra hircus*	(0,1)	(2,0)	(0,0)	(2,1)
4	*Bos taurus*	(0,1)	(2,0)	(0,0)	(2,1)
4	*Sus scrofa*	(1,2)	(1,1)	(0,1)	(2,4)
2	*Pteropus vampyrus*	(2,1)	(1,0)	(1,1)	(4,2)
2	*Pteropus alecto*	(2,1)	(1,0)	(1,1)	(4,2)
3	*Gallus gallus*	(6,0)	(3,0)	(1,0)	(10,0)
3	*Pelodiscus sinensis*	(7,0)	(2,0)	(1,0)	(10,0)
3	*Danio rerio*	(5,0)	(2,0)	(2,1)	(9,1)
3	*Salmo salar*	NA	NA	(3,1)	(3,1)

**Table 2 molecules-25-05906-t002:** Nineteen species and their associated ACE2 sequences.

Name	Species	ACE2 Accession ID	Length
S1	*Homo sapiens*	NP_001358344.1	805
S2	*Pan troglodytes*	PNI38577.1	805
S3	*Macaca mulatta*	XP_028697658.1	805
S4	*Equus caballus*	XP_001490241.1	805
S5	*Felis catus*	NP_001034545.1	805
S6	*Mesocricetus auratus*	XP_005074266.1	805
S7	*Manis javanica*	XP_017505752.1	805
S8	*Pteropus alecto*	XP_006911709.1	805
S9	*Capra hircus*	AHI85757.1	804
S10	*Sus scrofa*	NP_001116542.1	805
S11	*Mustela putorius furo*	XP_004758943.1	805
S12	*Bos taurus*	NP_001019673.2	804
S13	*Rattus norvegicus*	NP_001012006.1	805
S14	*Pteropus vampyrus*	XP_011361275.1	804
S15	*Rhinolophus ferrumequinum*	XP_032963186.1	805
S16	*Gallus gallus*	XP_416822.2	808
S17	*Pelodiscus sinensis*	XP_006122891.1	808
S18	*Danio rerio*	XP_005169417.1	807
S19	*Salmo salar*	XP_014062928.1	695

**Table 3 molecules-25-05906-t003:** Contact residues of RBD spike protein of SARS-CoV-2 and Homo sapiens ACE2.

SARS-CoV-2 RBD	ACE2 (Homo Sapiens)	SARS-CoV-2 RBD	ACE2 (Homo Sapiens)
K417	Q24	Q493	Q42
G446	T27	G496	L79
Y449	F28	Q498	M82
Y453	D30	T500	Y83
L455	K31	N501	N330
F456	H34	G502	K353
A475	E35	Y505	G354
F486	E37		D355
N487	D38		R357
Y489	Y41		R393

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
