# Peer review of "Possible Transmission Flow of SARS-CoV-2 Based on ACE2 Features"

_molecules, 2020, doi:10.3390/molecules25245906_

Round 1

Reviewer 1 Report

"I appreciated this manuscript by its conciseness. He does not get lost in methodological details and gets straight to the point. Of course, it asks for research of the associated literature if one needs methodological details, it is not a question here of bioinformatic methodology but of cell biology. This work aims to study the possibility of SARS-COV2 infecting different hosts (actual or potential) based on the main characteristics of the SARS-COV2 receptor ACE2. The choice of the 19 species is in accordance with the possible transmissibility of the virus to these species. The choice of Danio and Salmo, very far phylogenetically from the usual hosts, seems questionable. However, based on the structure of their ACE2, this makes them possible to show that the conservation of certain amino acid residues (or their substitution by aa of the same type) make transmissibility probable. The study compiles data on phylogeny, secondary structures, polarity. Each of these criteria have been studied using already existing methods and proven by the literature. It is not a question, in this study, but once again, of validating methods but of using robust methods to set up a precise investigation of the infectious potential of the virus between different taxa. Each given set permit the definition of clusters grouping together the closest species. The definition of these clusters shows a good control of the problematic associated with this type of work. This work certainly required a lot of attention and a lot of data verification.

The fact of mixing the different methods suggests that the results obtained are robust. Such a diverse approach is sometimes lacking in this type of manuscript. This seems to be an important point to emphasize here. Of course, this work could be done on more data. However, I doubt that many species (apart from "model" species) are already present in the databases.

The results allow us to conclude on the possible transmission of the virus within the clusters closest to humans. Unsurprisingly, these results conclude that, among the species studied, it is the members of primates that are the most at risk, followed by those of carnivores, certiodactyls and finally bats. It will be necessary to question the possibility for these different species of transforming themselves, in the long term, into healthy carriers of the virus or even into transmitters and diffusers of the disease. I hope to find other manuscripts that will dig in this direction in the comings days."

Author Response

We are very thankful to this reviewer for the thorough evaluation of our work. We completely agree that this work can be and should be extended further with the analysis of more ACE2 sequences from other species. These initial studies represent an important direction for the future development of this project, and we are planning to conduct these studies in the near future to investigate possible SARS-CoV-2 transmission to distant species. To further emphasize the importance of this recommendation, we added the following statement to the revised manuscript: “The results reported in this study allow us to propose possible routes of the SARS-CoV-2 transmission flow among various species. Unsurprisingly, our results indicate that, among the species studied, the members of primates are most at risk, followed by carnivores, cetartiodactyls, and finally bats. It is settling to see that the predicted transmission flow based on the results of our analyses is in line with the conventional evolutionary knowledge and reported infection cases. One should keep in mind though that the major goal of this study was to provide formally comprehensive structural evidence that could help in clarifying why some hosts are more susceptible than others to SARS-CoV-2 and could constitute a reservoir for further virus spillover. Obviously, more detailed studies are needed in the future to take into account structural properties of ACE2 and peculiarities of its interaction with the RBD of the S protein [15-18], the presence of different ACE2 isoforms in one animal species (e.g., humans have at least five ACE2 isoforms). Moreover, one should consider the epigenetic regulation and expression determination of ACE2 (e.g., despite having the same protein sequence, ACE2 is differently expressed in different human cells, and different levels of expression of ACE2 are found in the same type of nasal epithelial cells or pneumocytes from humans and mice). It will also be necessary to analyze more ACE2 sequences from other species and to investigate the possibility for these different species of transforming themselves, in the long term, into healthy carriers of the virus or even into transmitters and diffusers of the disease.

Reviewer 2 Report

The manuscript presents a bioinformatic study of ACE2 protein sequences cross 19 common vertebrate species. Therefrom, they predict potential susceptibility to SARS-CoV2 infection. The study provides extensive ACE2 sequence alignment, domain cluster and phylogenic analysis; however, hardly correlates to a real viral permissiveness test either in vitro or in vivo. The major concerns include:

  • Not consider the structural conformation of ACE2, and its interaction and affinity to RBD
  • Not consider different ACE2 isoforms in one animal species, such as human has at least five ACE2 isoforms
  • Not consider the epigenetic regulation and expression determination of ACE2. For example, despite having the same protein sequence, ACE2 are differently expressed in different human cells and different from humans and mice even of same type of nasal epithelial cells or pneumocytes.
  • The predicted transmission flow (figure 13) and animal clusters for SARS-CoV2 susceptibility, are already known based on conventional evolutionary knowledge and reported infection cases.
  • The study only repeats/extends one part (sequence comparison) of previously studies as shown in the following publications as some examples, but have not cited in the references:
  • Luan J, Lu Y, Jin X, Zhang L. Spike protein recognition of mammalian ACE2 predicts the host range and an optimized ACE2 for SARS-CoV-2 infection. Biochem Biophys Res Commun. 2020;526(1):165‐169.
  • Qiu Y, Zhao YB, Wang Q, et al. Predicting the angiotensin converting enzyme 2 (ACE2) utilizing capability as the receptor of SARS-CoV-2 [published online ahead of print, 2020 Mar 19]. Microbes Infect. 2020;S1286-4579(20)30049-6.
  • Sang ER, Tian Y, Gong Y, Miller LC, Sang Y. Integrate structural analysis, isoform diversity, and interferon-inductive propensity of ACE2 to predict SARS-CoV2 susceptibility in vertebrates. Heliyon. 2020 Sep;6(9):e04818.
  • Sun J, He WT, Wang L, et al. COVID-19: Epidemiology, Evolution, and Cross-Disciplinary Perspectives. Trends Mol Med. 2020;26(5):483‐495.

Author Response

The manuscript presents a bioinformatic study of ACE2 protein sequences across 19 common vertebrate species. Therefore, they predict potential susceptibility to SARS-CoV2 infection. The study provides extensive ACE2 sequence alignment, domain cluster and phylogenetic analysis; however, hardly correlates to a real viral permissiveness test either in vitro or in vivo. The major concerns include:

1. Not consider the structural conformation of ACE2, and its interaction and affinity to RBD

  1. Not consider different ACE2 isoforms in one animal species, such as humans have at least five ACE2 isoforms

  2. Not consider the epigenetic regulation and expression determination of ACE2. For example, despite having the same protein sequence, ACE2 are differently expressed in different human cells and different from humans and mice even of the same type of nasal epithelial cells or pneumocytes.

Response: We are thankful to the reviewer for careful reading of the manuscript. The critiques are well taken, however, we would like to emphasize here that the main scope of the paper was not to provide a functional dissection, in biological terms, of the subtle differences that could account for susceptibility to SARS-CoV-2 or virus carrier state of different animal species, such as ACE2 isoforms or epigenetic control of the ACE2 expression. Indeed we intended to pave the way to such studies by simply providing formally comprehensive structural evidence that could help in clarifying why some hosts are more susceptible than others to SARS-CoV-2 and could constitute a reservoir for further virus spillover. We fully agree that the experiments you indicated are certainly warranted and must be pursued in the near future. Therefore we have included this point in our revised discussion.  The last part of the discussion section now reads: “The results reported in this study allow us to propose possible routes of the SARS-CoV-2 transmission flow among the species. Unsurprisingly, our results indicate that, among the species studied, it is the members of primates that are the most at risk, followed by those of carnivores, cetartiodactyls, and finally bats. It is settling to see that the predicted transmission flow based on the results of our analyses is in line with the conventional evolutionary knowledge and reported infection cases. One should keep in mind though that the major goal of this study was to provide a formally comprehensive structural evidence that could help in clarifying why some hosts are more susceptible than others to SARS-CoV-2 and could constitute a reservoir for further virus spill over. Obviously, more detailed studies are needed in the future to take into account structural properties of ACE2 and peculiarities of its interaction with the RBD of the S protein [15-18], and the presence of different ACE2 isoforms in one animal species (e.g., humans have at least five ACE2 isoforms [26]). Moreover, one should consider the epigenetic regulation and expression determination of ACE2 (e.g., despite having the same protein sequence, ACE2 is differently expressed in different human cells, and different levels of expression of ACE2 are found in the same type of nasal epithelial cells or pneumocytes from humans and mice ). It will be also necessary to analyze more ACE2 sequences from other species and to investigate the possibility for these different species of transforming themselves, in the long term, into healthy carriers of the virus or even into transmitters and diffusers of the disease.”

  1. The predicted transmission flow (figure 13) and animal clusters for SARS-CoV2 susceptibility, are already known based on conventional evolutionary knowledge and reported infection cases.

Response: We are glad that the predicted transmission flow based on the results of our analyses is in line with conventional evolutionary knowledge and reported infection cases.

  1. The study only repeats/extends one part (sequence comparison) of previously studies as shown in the following publications as some examples, but have not cited in the references:

Luan J, Lu Y, Jin X, Zhang L. Spike protein recognition of mammalian ACE2 predicts the host range and an optimized ACE2 for SARS-CoV-2 infection. Biochem Biophys Res Commun. 2020;526(1):165‐169.
Qiu Y, Zhao YB, Wang Q, et al. Predicting the angiotensin converting enzyme 2 (ACE2) utilizing capability as the receptor of SARS-CoV-2 [published online ahead of print, 2020 Mar 19]. Microbes Infect. 2020;S1286-4579(20)30049-6.
Sang ER, Tian Y, Gong Y, Miller LC, Sang Y. Integrate structural analysis, isoform diversity, and interferon-inductive propensity of ACE2 to predict SARS-CoV2 susceptibility in vertebrates. Heliyon. 2020 Sep;6(9):e04818.
Sun J, He WT, Wang L, et al. COVID-19: Epidemiology, Evolution, and Cross-Disciplinary Perspectives. Trends Mol Med. 2020;26(5):483‐495.

Response: Suggested references have been added in the revised manuscript.

Reviewer 3 Report

The Authors performed a comprehensive bioinformatics analysis of the ACE2 protein of nineteen species that allowed a prediction of the interspecies SARS-CoV-2 transmission. The work is done very carefully.  The results are thoroughly discussed. This is a valuable work which provide a very interesting and needed results.

Author Response

The Authors performed a comprehensive bioinformatics analysis of the ACE2 protein of nineteen species that allowed a prediction of the interspecies SARS-CoV-2 transmission. The work is done very carefully.  The results are thoroughly discussed. This is a valuable work which provides very interesting and needed results.

Response: We are thankful to this reviewer for commending our work, and greatly appreciate time spent on evaluation of our manuscript.  

Round 2

Reviewer 2 Report

The revised manscript has addressed some of my previous concerns using discussion but little data supplement. Further improvement in scientific robustness, especially protein structure information are recommended. The fact is that the SARS-CoV2 susceptibility can not be simply determined by the primary ACE2 sequence similarity. 

Author Response

The revised manscript has addressed some of my previous concerns using discussion but little data supplement. Further improvement in scientific robustness, especially protein structure information are recommended. The fact is that the SARS-CoV2 susceptibility can not be simply determined by the primary ACE2 sequence similarity.

REPLY: Thank you for pointing this out. To address this concern, we added discussion of some structural aspects of ACE2 proteins and two new figures to the revised manuscript. The following section is included to the discussion section of the manuscript:

“It is well known that protein pairs with a sequence identity higher than 35–40% are very likely to be structurally similar [29–31], whereas protein pairs with a sequence identity of 20–35% represent a ‘twilight zone’, where structural similarity in pairs is considerably less common, with less than 10% of protein pairs with sequence identity below 25% have similar structures [29–31]. Therefore, since sequence identity of the ACE2 proteins from nineteen species analyzed in this study ranges from 99.01% (Homo sapiens vs. Pan troglodytes) to 58.0% (Homo sapiens vs. Danio rerio), with the lowest identity of 57.13% being between the proteins from Danio rerio and Rhinolophus ferrumequinum, one might expect rather close overall structural organization of all these proteins, even the most distant ones. In fact, even the lowest level of sequence identity for the pair of ACE2 proteins is still well above the sequence identity of 20–35% characteristic for the ‘twilight zone’. On the other hand, fold-level, global structural similarity does not exclude the presence of local structural variability that might define, for example, the peculiarities of protein-protein interactions. Structural information is currently available only for the ACE2 from Homo sapiens and Felis catus. Therefore, previous studies that analyzed the peculiarities of interactions between the viral spike protein and host ACE2 from many household and other animals, such as Pan troglodytes (chimpanzee), Macaca mulatta (Rhesus monkey), Felis catus (domestic cat), Equus caballus (horse), Oryctolagus cuniculus (rabbit), Canis lupus familiaris (dog), Sus scrofa (pig), Avis aries  (sheep), Bos taurus (cattle), Mus musculus (house mouse), and Mustela putorius furo (ferret) [32–34] were focused on the structural part of these interactions and utilized a typical set of structural biology approaches, such as homology modelling and docking. Therefore, in line with our previous study [25], we decided to compare the peculiarities of the per-residue intrinsic disorder predispositions of the ACE2 proteins from nineteen species analyzed in this study rather than building their homology models. Fig.14 summarizes the results of this analysis and shows that although these proteins have rather similar intrinsic disorder predispositions, their disorder profiles are not identical. Furthermore, such differences in the intrinsic disorder predisposition are not equally spread through the protein sequences, with some regions (e.g., the N-terminal 150 residues and residues 500-700) of the disorder profiles showing rather noticeable variability. Fig.14 also shows that the S protein binding domains D1 and D2 of ACE2 proteins are characterized by high variability of their intrinsic disorder predispositions, whereas D3 domains are more conserved. We also looked at the peculiarities of intrinsic disorder profiles of ACE2 proteins in six clusters with the major focus at the S protein binding domains D1, D2, and D3 (see Fig.15). This comparison revealed that as a rule, the in-cluster variability of intrinsic disorder propensity was noticeably lower than the diversity between the clusters. These observations support the notion that the capability of ACE2 to interact with SARS-CoV-2 protein S can be dependent on the peculiarities of the ACE2 local intrinsic disorder predisposition [25].”